# A versatile cryo-transfer system, connecting cryogenic focused ion beam sample preparation to atom probe microscopy

Chandra Macauley [1,2] *, Martina Heller[1,2], Alexander Rausch[1], Frank Kümmel[1]¤, Peter Felfer[1]

**1** Institute I, Materials Science & Engineering Department, Friedrich-Alexander-Universität Erlangen-Nürnberg (FAU), Erlangen, Germany, **2** Interdisciplinary Center for Nanostructured Films (IZNF), Erlangen, Germany

¤ Current address: Heinz Maier-Leibnitz Zentrum (MLZ), TU München, Garching, Germany
* chandra.macauley@fau.de

**Data Availability Statement:** All relevant data are within the manuscript and its Supporting Information files. The APT file is available on

## Abstract

Atom probe tomography (APT) is a powerful technique to obtain 3D chemical and structural information, however the 'standard' atom probe experimental workflow involves transfer of specimens at ambient conditions. The ability to transfer air- or thermally-sensitive samples between instruments while maintaining environmental control is critical to prevent chemical or morphological changes prior to analysis for a variety of interesting sample materials. In this article, we describe a versatile transfer system that enables cryogenic- or room-temperature transfer of specimens in vacuum or atmospheric conditions between sample preparation stations, a focused ion beam system (Zeiss Crossbeam 540) and a widely used commercial atom probe system (CAMECA LEAP 4000X HR). As an example for the use of this transfer system, we present atom probe data of gallium- (Ga)-free grain boundaries in an aluminum (Al) alloy specimen prepared with a Ga-based FIB.

## 1. Introduction

Cryogenic sample preparation and characterization has been developed and widely used in the biology community since the 1970s [1] and is only recently being applied more broadly to materials science. These techniques open up a wide range of materials, including soft materials and environmentally unstable materials, that can now be studied e.g. in electron microscopes such as transmission electron microscopes (TEMs) [2, 3] and scanning electron microscopes (SEMs) [4, 5]. Such cryogenic/environmental sample preparation techniques could also open up new research fields when applied to atom probe tomography (APT), an increasingly popular, high-resolution microscopy / mass spectrometry technique that is capable of determining the chemistry and structure of materials in 3-dimensions with single atom sensitivity [6]. Although conductive, non-site specific samples are relatively easily prepared with electropolishing [7], preparation of non-conductive or site-specific APT samples is almost always achieved using a combined focused ion beam/scanning electron microscope (FIB/SEM) to

Figshare: https://figshare.com/articles/dataset/R56_02188-Cryo_Al_pos/12895898.

**Funding:** C.M., M.H. and P.F. acknowledge financial support by the Bavarian Ministry of Economic Affairs and Media, Energy and Technology for the joint projects in the framework of the Helmholtz Institute Erlangen-Nürnberg for Renewable Energy (IEK-11) of Forschungszentrum Jülich. The authors would also like to acknowledge funding by the Deutsche Forschungsgemeinschaft (DFG) via the Cluster of Excellence 'Engineering of Advanced Materials' (project EXC 315). The funders provided support in the form of salaries for authors CM and MH, but did not have any role in study design, data collection and analysis, decision to publish, or preparation of the manuscript.

**Competing interests:** The authors have declared that no competing interests exist.

make samples with the required geometry [8–11]. Conventional transfer of samples from the FIB/SEM to the atom probe takes place at ambient temperature and pressure, causing chemical or morphological changes or even complete destruction of environmentally- or thermally-sensitive samples. Therefore, a system that enables cryogenic- and/or environmental-transfers is critical to broadening the research fields to which APT can be applied.

Some cryogenic- and environmental-transfer systems are commercially available and rely on a specimen shuttle transfer device or suitcase that docks with various instruments and maintains a user-defined environment during transfer [12–15]. These systems can be broadly separated based on if the transfer devices are a) actively cooled and pumped e.g. Ferrovac's cryo suitcase [15, 16], b) only actively cooled e.g. Leica's VCT500 [17, 18] and c) completely passive e.g. Quorum's PP3006 CoolLok [14]. There are advantages and disadvantages to each of these systems. While the actively cooled and pumped Ferrovac Ultra High Vacuum (UHV) Cryo Transfer Module (CTM) provides the most controlled transfer conditions (vacuum below $10^{-9}$ mbar, temperature down to -180˚C), the additional hardware required increases the size and weight of a liquid nitrogen-filled transfer device to ~1 m and 14 kg [16]. Furthermore, the Ferrovac suitcase requires a free CF40 port, which may not be available on well-equipped microscopes. On the other hand, the passive transfer systems are often more compact but require quick transfer times between instruments to minimize sample degradation. Despite the need for quick transfers, passive systems have successfully been used to observe hydrogen trapping in steels [19] and intact organic molecules [14]. A recent publication by McCarroll et al. provides an overview of some of the most common systems [20].

In this paper, we describe a custom-built versatile transfer system that enables transfer of samples that are sensitive to air or thermal exposure between sample preparation stations such as a FIB/SEM and a local electrode atom probe (LEAP). The paper is organized in the order of components/modifications used during the cryo-FIB preparation of an atom probe sample and subsequent transfer into the LEAP. Finally, to demonstrate the actively cooled cryogenic-, high vacuum-transfer capabilities of the system, an aluminum (Al) alloy was milled using a gallium (Ga) liquid-metal ion source FIB at cryogenic temperatures prior to being transferred using the described system to the LEAP. While Al alloys are not per-se environmentally sensitive, the exposure to Ga in general [21], and in the FIB in particular [22–24], leads to strong segregation of the Ga to the grain boundaries, in a process called liquid metal embrittlement. The altered chemical makeup of the grain boundary is then no longer representative of the original specimen [25]. However, the site-specific investigation of the grain boundaries in these Al alloys is of great interest, as they define the susceptibility to intergranular corrosion [26–28], a major issue in their use e.g. in the automotive industry [29]. As shown below, this problem can be circumvented by the use of low temperatures throughout the characterization and transfer process.

While the ability to quickly make environmentally controlled transfers between instruments has been shown by other groups, our system has unique advantages such as a small footprint and close shielding of the sample that prevents frost formation. Since it uses standard parts or components that can easily be milled, the system is highly adaptable to new experimental arrangements. Additionally, the system does not require a dedicated cryo port on the FIB/SEM, as it interlocks directly with the existing load lock door, a feature that is ideal for multi-user instruments that are not exclusively used for cryo experiments.

## 2. Materials and methods

In the course of this work, we have established a cryogenic route connecting a Zeiss Crossbeam 540 FIB/SEM (Carl Zeiss, Oberkochen, Germany), equipped with a Quorum cryogenic

PP3005 SEMCool (Quorum Technologies, Bath, UK) cold stage, with a CAMECA 4000X HR local electrode atom probe (LEAP) system through a cryogenic transfer system designed by the authors. A general overview of the cryogenic/vacuum transfer system including relevant temperatures and pressures at the various steps of sample preparation, transfer and analysis is shown in Fig 1. Fig 1A, a flow chart of the preparation process is given, including approximate transfer times. All steps colored in blue indicate the use of new components designed and built in this work. Fig 1B gives a schematic overview of how the components interface with the FIB/SEM and LEAP. The sample is first transferred into the FIB/SEM analysis chamber, AC, using a custom transfer device (FIBTD) that attaches directly to the existing load lock, LL, Fig 1B, top. After FIB milling and/or SEM characterization, the sample is moved out of the FIB/SEM using the same, thermally-insulated, but not actively cooled FIBTD, in which the sample is handed off to an actively cooled transfer device (ACTD), with a standard KF16 vacuum flange (Fig 1B, center). The ACTD is used to transport the sample to the LEAP and attaches to a custom 'cryo' load lock, that allows for direct insertion of the sample into the LEAP's buffer chamber, BC (Fig 1B, bottom). In the buffer chamber, a special specimen shuttle puck is waiting pre-cooled in a liquid nitrogen cooled insertable stage. The transfer rod in the LEAP is then used to move the sample onto the stage in the analysis chamber. In the cold chain, the sample can be kept at cryogenic temperatures at all times ($<$ ca. - 160˚C), so long as the hand-off times between systems, when the sample is not actively cooled, are kept low (some 10s of s, see S1 File). This was confirmed with temperature measurements shown in S1 and S2 Figs and S1

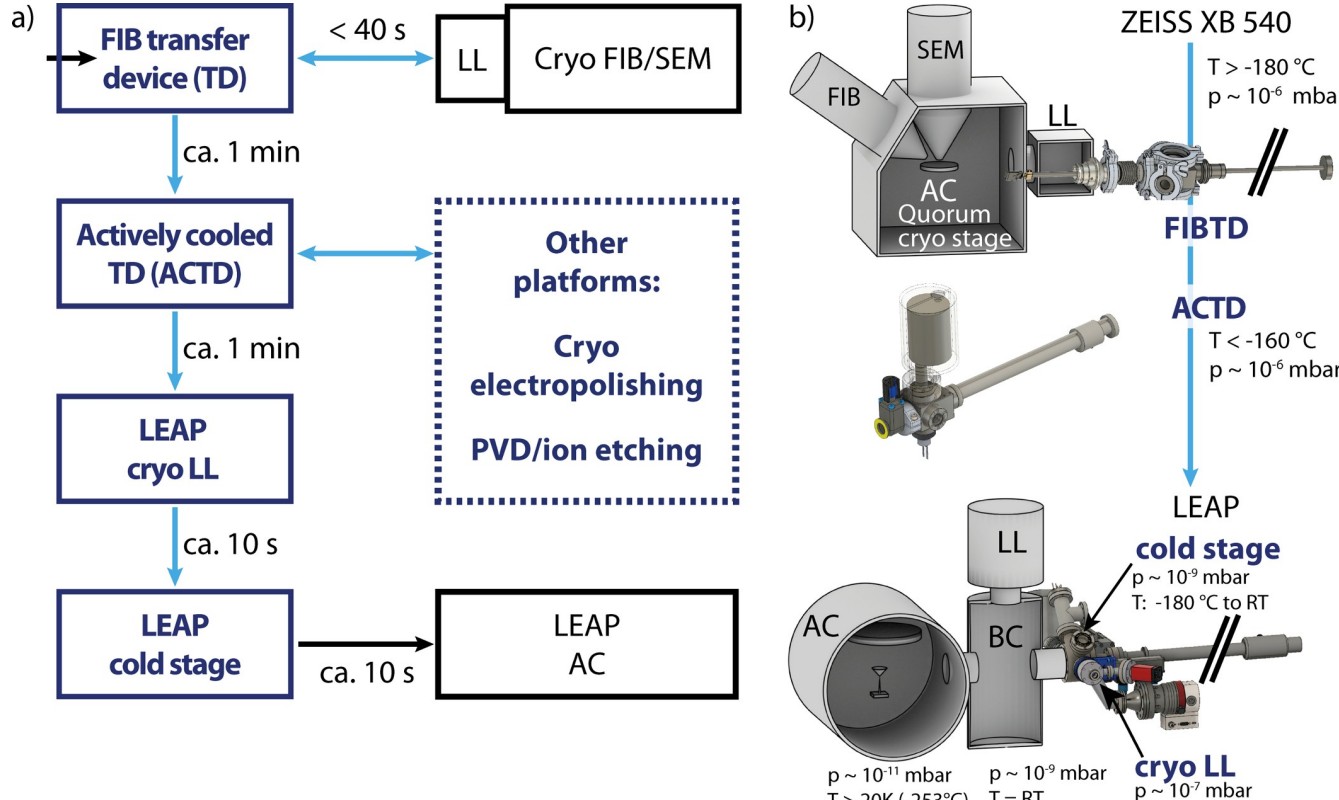

**Fig 1. An experimental overview of the transfer system.** In a) the workflow is shown with transfer times. Custom components designed and built as part of this work are blue while existing systems are black A schematic overview in b) shows the FIB transfer device connected directly to a Zeiss Crossbeam 540 FIB/SEM. Prepared samples can be transferred with an actively cooled transfer device (ACTD) that interfaces with existing instrument load locks (LL) into a CAMECA LEAP. AC: analysis chamber, BC: buffer chamber, RT: room temperature.

Table in S1 File. Here we use the phrase "cold chain" to describe the protocol of transferring samples from the FIB/SEM to the LEAP while maintaining cryogenic temperatures. The ACTD interfaces with other instrumentation through an industry standard KF16 flange, making it easily adaptable to multiple sample preparation and analysis stations. Although not discussed in detail in this paper, the ACTD connects to other platforms, shown in the dashed box in Fig 1A, such as a cryo electropolishing station and PVD coating/ion etching station. Photos of these systems are shown in S3 and S4 Figs in S1 File, respectively. Each component required for the extension of the commercial systems to cryo-capability was designed and built in-house, using standard vacuum parts whenever possible. Custom components were fabricated either by the Mechanics and Electronics Workshop of the Faculty of Engineering at Friedrich-Alexander-University (Erlangen, Germany) or using a miniature 5-axis CNC mill (Pocket NC, Bozeman, MT, USA).

## 2.1 Cryogenic FIB/SEM stage and transfer device

The basis of the sample preparation in this work is the FIB/SEM. In our instrument, samples can be kept at any temperature from room temperature down to around -190˚C, by using a Quorum PP3005 cryogenic stage. This stage is cooled by a stream of gaseous nitrogen, which is passed through liquid nitrogen in a heat exchanger and thus brought to near liquid nitrogen temperatures. Temperature regulation is achieved by adjusting the cold gas flowrate for coarse tuning in combination with a heating element for closed-loop temperature control. This thermally insulated cold stage attaches to the existing SEM stage, as shown in Fig 2A. This figure shows an APT sample mounted in a double threaded APT sample carrier ('double nipple') which is in turn mounted in a FIB cryo-shuttle.

Custom, pre-tilted (54˚, which is the angle between the electron and ion beams) FIB cryo-shuttles with threads to accommodate the specimens were made to allow annular ion beam milling perpendicular to the sample surface without tilting the stage. This is useful since the cryo-stage is supplied with cooling gas through polymer tubes which get stiff at cold temperatures. While tilting at cryogenic temperatures is possible, it needs to be carried out carefully to not damage the gas lines or stage. By using pre-tilted samples, the ion beam can easily be used to cut into materials or sharpen tips into the required geometry for atom probe tomography. If

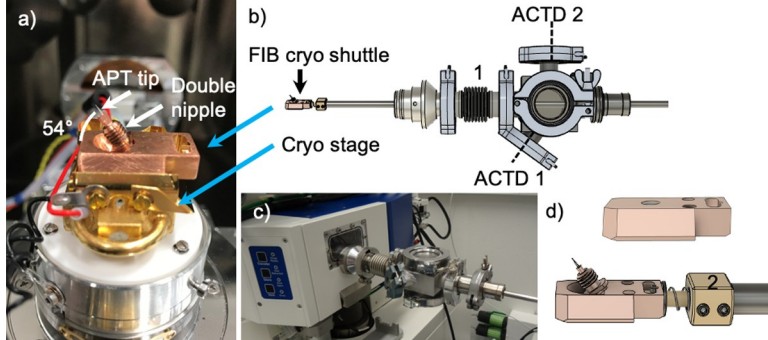

**Fig 2. The cryo stage and FIB transfer device (FIBTD) enable characterization, sample preparation and transfer at cryogenic conditions.** a) A custom 54˚ pre-tilted FIB cryo-shuttle mounted on the Quorum cryo-stage with key components and directions labeled. b) Rendered CAD drawing of the FIBTD indicating the flexible bellows coupling, 1, and the two possible connection points for the actively cooled transfer device (ACTD). c) The FIBTD connects directly to the existing FIB load-lock. d) The non-pre-tilted (top) and 54˚ pre-tilted FIB cryo-shuttles (bottom) attach to the end of the FIBTD with a bayonet connection and are thermally-isolated from the transfer rod with a polyether ether ketone end-component, 2.

perpendicular FIB milling is not required for a given experiment, but SEM imaging and cryo- or environmental-transfer to other systems is desired, non-pre-tilted FIB cryo-shuttles can be used as shown in Fig 2B.

To facilitate quick cryogenic- or environmental-transfers to/from the FIB/SEM, a transfer device was designed and built, as shown in Fig 2B. This FIB transfer device (FIBTD) interfaces with the existing Zeiss 80 mm Airlock without further modification, as shown in Fig 2C. It also allows for a hand-off to the actively cooled transfer device (ACTD), for FIB cryo-shuttles with pre-tilt ('ACTD1' in Fig 2B) and without pre-tilt ('ACTD2' in Fig 2B). The transfer device reduces the risk of chamber contamination and enables efficient characterization of multiple samples in a single FIB session, since without the means to transfer cold samples through the load-lock, the cold stage would need to be warmed to room temperature for sample exchanges and the entire analysis chamber vented. While commercial load-lock solutions exist that attach to the ports of the chamber, no free ports to attach such a solution were available in our current multi-user FIB/SEM instrument, which is not exclusively used for cryogenic characterization. To prevent unwanted vibrations in the FIB/SEM and to clear the view of the instrument opera- tor, the FIBTD is easily removed after each transfer.

For a sample exchange, the existing Airlock roughing pump is used to pump down the air- lock with the attached FIBTD prior to introducing the sample to the pre-cooled stage. This pumping operation takes ca. 30s until a vacuum sufficient for transfer is established. The gen- erated vacuum of $\sim 10^{-4}$ mbar thereby has proven to be sufficient to prevent frost built up on the specimen, as shown below in the analysis of the Al alloy. This is likely also a result of using dry nitrogen as a flushing gas for the load lock, keeping the $H_2O$ partial pressure low. As the cryo-stage is ~18mm taller than the stock SEM stage, the stage has to be lowered to its mini- mum Z-height for sample exchange. A flexible bellows coupling (Figs 2B and 1) is used to allow the FIBTD to pivot slightly upwards, so the FIB cryo-shuttle on the FIBTD with the cold stage. A bayonet connection to the specimen shuttle (Fig 2D) is used to hold the shuttle at the end of the transfer rod securely, thus ensuring rapid, reliable transfers. In this FIBTD, no active cooling or pumping is available. The sample is, however, thermally insulated from the rest of the transfer device via a low thermal conductivity polyether ether ketone (PEEK) end-effector, 2 in Fig 2D, to minimize heat transfer. To measure the efficacy of this thermal isolation, speci- men temperature was measured while a FIB cryo shuttle, pre-cooled by the cryo stage to -118˚C, warmed up (since it was not actively cooled) during a simulated hand-off to the ACTD. As shown in S2 Fig in S1 File, the pre-cooled shuttle on the PEEK end-effector of the FIBTD warms up at 2.3˚C/min at a pressure below $10^{-5}$ mbar. The importance of pressure on the warming rates of pre-cooled FIB cryo shuttles attached to the FIBTD transfer rod is dem- onstrated in S1 Table in S1 File.

## 2.2 Actively cooled transfer device (ACTD)

To facilitate transport of the samples between the FIB/SEM and the LEAP, an actively cooled transfer device (ACTD) was developed. The ACTD can be cooled using liquid nitrogen or potentially other cryogenic liquids via a cold finger, while temperature and pressure are moni- tored and logged to alert if all the liquid nitrogen is used up or a vacuum leak occurs. So long the reservoir does not run dry, the sample is can be as cold as -170˚ C in the ACTD. Active pumping through an ion getter pump is also possible but has so far not been used due to the short transfer times between instruments (< 10 minutes) within the facility. Active pumping would only be necessary for longer transfer times > 15 min, e.g. between institutions. The ACTD is shown in Fig 3A. Due to the small size of the atom probe samples, the ACTD can be small and compact, (41 cm long, 2 kg), and can easily be carried in one hand, while still

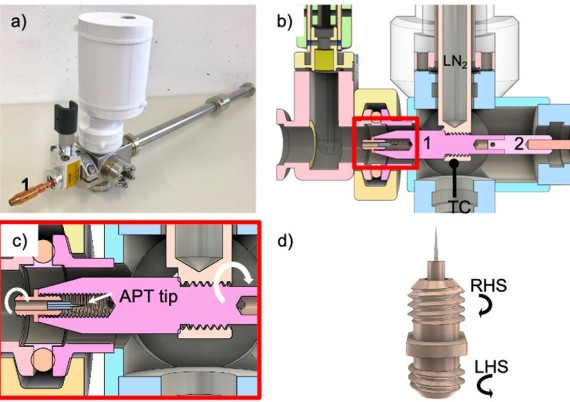

**Fig 3. The actively cooled transfer device (ACTD) and the double nipple sample shuttle enable transfers to the atom probe.** a) Photo of the ACTD showing the valve open and the copper end-component (labeled 1) extended. b) The rendered cross-section of the ACTD shows the threaded connection between the liquid nitrogen (LN$_2$) cold finger and the copper end-component (1). A low thermal conductivity PEEK component (2) thermally isolates the end-component from the rest of the transfer rod. Temperature is measured at the cold finger using a thermocouple (TC). c) A higher magnification inset of b) more clearly shows how the double nipple screws into the copper end-component during transfers, resulting in the atom probe tip being shielded. d) Transfers between the FIBTD and ACTD are enabled by the two-threaded double nipple that has a right-handed screw (RHS) thread and a left-handed screw (LHS) thread.

allowing for bottom-out pressures in the range of $10^{-8}$ mbar through its almost entirely metal sealed construction with a magnetically-coupled transfer rod. These pressures are reached after a pump down of several hours to a day, depending on ambient humidity, if the ACTD was exposed to air. The main body of the ACTD is machined from a solid piece of 316 stainless steel, with welded-in CF 16 flanges and a KF 16 flange to the gate valve. As a result, the lowest achievable pressure is dictated by the gate valve, which is a KF 16 gate valve with aluminum body and rubber gasket sealed flanges (VAT Valves, Switzerland). Although these KF valves have the same leak tightness as their CF metal flanged, fluoroelastomer (FKM)-sealed counterparts, they cannot be heated beyond 80˚C, limiting possible bake-outs. Such higher temperature bake-outs are only needed to reach ultra high vacuum ($<10^{-9}$ mbar) conditions. Since the ACTD is being connected to the FIB/SEM, which has an ultimate pressure of $10^{-7}$ mbar, such bake-outs are not required.

Not only does the small size of the ACTD aid in the ease of transfers, it also minimizes frost contamination through its comparably small volume. As seen in the photograph and computer aided design (CAD) cross-section drawing in Fig 3A and 3B, respectively, a 210 mL liquid nitrogen-filled steel vessel provides active cooling via a direct threaded connection between the copper end-component and the spout of the vessel, enabling temperatures of -160˚C during transfer. A detailed inset is provided in Fig 3C to more clearly show how the atom probe sample is secured in the ACTD. To protect the user from cold metal parts, the steel vessel is surrounded by a 3D-printed plastic hull. The sample is held in a double-nipple, Fig 3D, that has a 5 mm diameter right-handed screw (RHS) thread on one side, and a left-handed screw (LHS) thread on the other. Such a design enables the double nipple to be transferred between the copper end-component shown in Fig 3A, and FIB cryo shuttle or LEAP shuttle pucks with one continuous twisting motion, as will be described in the next section. The sample itself can be a rough-electropolished tip crimped in a copper tube or a half-TEM grid that contains several tips. With a clear bore of a minimum 16mm, larger sample diameters could be accommodated if needed. During transfer, the sample is closely shrouded by the copper end-component (5 mm inner diameter), as shown in the higher magnification cross-section in Fig 3C,

minimizing frost formation. In this way, as the sample is screwed into the transfer rod, any contaminant will condense on the outside of the copper sheath rather than on the sample. The copper end-component is thermally isolated from the rest of the aluminum transfer rod with a low thermal conductivity plastic (PEEK) component. Temperature is measured using a K-type thermocouple (TC in Fig 3B) in contact with the liquid nitrogen (LN$_2$) cold finger, since the rotating motion of the transfer rod does not allow for direct attachment to the sample or the end-effector. The pressure in the ACTD is monitored using a MEMS Pirani gauge (MKS instruments type 925), with a bottom-out pressure of 1x10$^{-5}$ mbar. A manually operated mini-gate valve (KF 16, VAT Group AG) isolates the ACTD and enables connection to the FIB transfer device, other sample preparation stations (see S1 File), and to the LEAP.

## 2.3 Hand-off between the FIBTD and the ACTD

The FIBTD must connect with the ACTD to move cryogenically-prepared, or otherwise environmentally-sensitive samples to the LEAP. The ACTD can connect with the 54˚-angled flange, as shown in the CAD drawing in Fig 4A, or if the sample is not mounted on a pre-tilted cryo-shuttle it can connect to the perpendicular flange (ACTD 2 in Fig 4A). Fig 4B shows both the FIBTD and the ACTD connected to the FIB/SEM load-lock. In this configuration, a (usually) pre-pumped ACTD is connected to the FIB transfer arm that is at ambient pressure. The FIB transfer device is then pumped down with the FIB/SEM load-lock, and the gate valve of the ACTD is opened. At this point, the vacuum is maintained by the turbo pump of the FIB/SEM system, resulting in a pressure of $< 10^{-5}$ to $10^{-6}$ mbar, as measured by the wide-range gauge of the FIB/SEM, which is significantly lower compared to moving the specimen into the FIB/SEM, where initially only the FIB/SEM load lock roughing pump maintains the vacuum. When the FIB cryo-shuttle is first removed from the cryo stage, Fig 4C, it must be rotated 90˚ along the axis of the FIB transfer rod, so that the pre-tilted double nipple and atom probe sample are parallel to the axis of the ACTD, Fig 4D. The threaded copper sheath of the ACTD is then screwed on to the double nipple, enshrouding the sample, Fig 4E. Thanks to the two-threaded design of the double nipple, the same twisting direction is used to first screw the copper sheath onto the double nipple before the double nipple is unscrewed from the FIB cryo

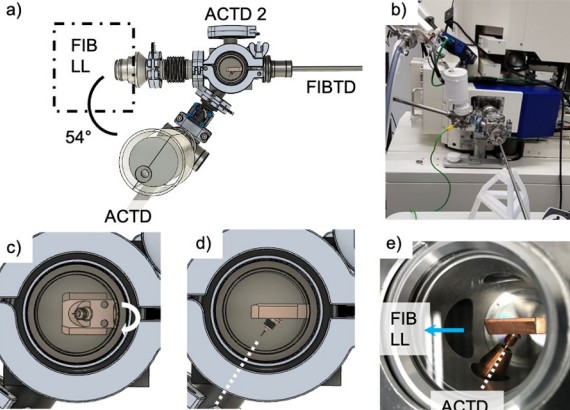

**Fig 4. Samples are handed-off between the FIBTD and the ACTD while the FIBTD is connected to the FIB load lock (LL).** a) CAD drawing depicting the 54˚ connection between the FIBTD and the ACTD. The perpendicular port for the ACTD that is labeled ACTD 2 can be used with the FIB cryo shuttles that are not pre-tilted. b) The connected transfer devices interface directly to the Zeiss load lock. When the FIB cryo shuttle is first removed from the FIB, c), it must be rotated 90˚ along the axis of the FIBTD transfer rod to line-up with the ACTD as shown in d). The copper end-component of the ACTD then surrounds the sample, e) as the double nipple is passed from the FIBTD to the ACTD.

shuttle, as shown in the movie provided in the S1 File. The ACTD transfer rod is then retracted, the gate valve is closed and the ACTD is transported to the LEAP while maintaining vacuum conditions below the measurement limit of the attached Pirani gauge ($10^{-5}$ mbar), simply by cryo-sorption for the relatively short time (ca. 5 min.) it takes to be transported from the FIB/SEM to the LEAP in our facility. This efficacy of this method has not yet been tested at another facility that requires much longer transfer times. During the exchange process, the sample is always connected to cold parts of either the FIBTD or the ACTD, so no temperature rise occurs (for temperature / pressure data from the hand-off see S1 File).

## 2.4 Modifications to the LEAP

In order to allow the samples to be inserted into the atom probe without breaking the vacuum / cold chain, modifications to our atom probe were also needed. To this end, we implemented a solution that enables direct transfer of the samples onto the LEAP transfer rod that moves samples into the main analysis chamber. The sample transfer takes place in what is commonly referred to as the buffer chamber of the LEAP, after which the sample can be quickly transferred into the analysis chamber. This solution consists of a 70mm long cryo-transfer chamber, labeled 1 in the CAD drawing in Fig 5A and 5D, which was installed to the right of the existing buffer chamber. This necessitated the installation of a slightly longer LEAP transfer rod (609 mm) in order to accommodate the additional length. The flange also contains a viewport for specimen alignment (2) as well as two other flanges (3) that accommodate: 1) a movable stage, highlighted with the square in Fig 5A and shown in detail in 5B and 5C, capable of in-situ heating and cryogenic cooling, and 2) the stage's electrical connections. The stage can be cooled with liquid nitrogen via a vessel (4 in Fig 5) that is also connected to the linear drive of the stage. When not in use, the stage can be retracted (Fig 5C, top), preventing interference with the normal use of the LEAP transfer rod. Prior to a cryo transfer, a custom LEAP shuttle puck is attached to the LEAP transfer rod, which is retracted fully before inserting the cryo stage, Fig 5C, bottom. The LEAP puck can then be placed on the cryo-stage to pre-cool before a cryo transfer is initiated. The ACTD connects directly to a flexible bellows coupling welded onto the cryo LL (5 in Fig 5). A 80 l/s turbo pump, as part of a pumping station configuration (Pfeiffer HiCube 80 Eco, 6) pumps down the volume of pipe between the ACTD and a VAT gate valve (7) while the pressure is measured by a combined Pirani and cold cathode vacuum gauge (8). When the pressure inside the pipe reaches a desired value, usually $10^{-6}$ mbar, the mini

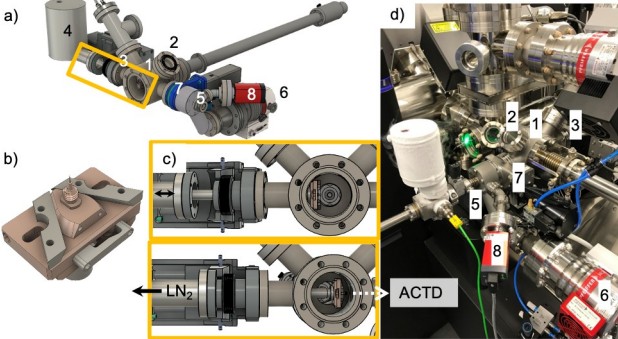

**Fig 5. Modifications were made to a LEAP to enable cryo/environmental transfers.** a) The rendered CAD drawing indicates the modifications to the LEAP, described in the text. A detailed view of the cryogenic-stage, is shown in b) with a custom LEAP shuttle in place as it would be during the hand-off of the sample and double nipple from the ACTD. The region highlighted with an orange box in a) is shown in c) to illustrate the movement of the cryo-stage from the perspective of the analysis chamber. d) A photo shows in detail the connection of the ACTD to the LEAP.

gate valve of the ACTD is opened. Once the pressure inside the ACTD is better than $5 \times 10^{-6}$ mbar, the gate valve to the LEAP buffer chamber is opened and the sample, which was cooled throughout the process with liquid nitrogen, can be screwed into the pre-cooled, custom LEAP puck, Fig 5B. During this hand-off, the LEAP puck is held in place by the actively cooled cryo stage, Fig 5C, bottom. A video of the complete cryo-transfer into the LEAP is provided in the S1 File. Once the hand-off is complete, the ACTD transfer rod is retracted and the cryo load lock VAT gate valve is closed. The LEAP transfer rod is used to remove the custom LEAP puck from the cryo stage, which is then retracted, and the puck can be moved into the LEAP analysis chamber. Since all of the additional equipment is mounted outside the regular buffer chamber of the LEAP instrument and the ACTD is easily removed when not in use, normal operation by regular users is unimpeded.

## 2.5 Utilization of the above cryo-transfer system to study Al alloys

To demonstrate the efficacy of the cryogenic-, environmental-transfer capabilities of the system, we chose to show the prevention of FIB-induced Ga segregation to the grain boundaries in the Al–Ga system. Normally, if Al is exposed to Ga, severe segregation of Ga to the grain boundaries of the Al is observed, as was discussed in the introduction. At low temperatures, we hypothesized that this phenomenon could be suppressed by inhibiting Ga diffusion.

To provide a sufficient chance of encountering one or more grain boundaries in the experiment, ultrafine-grained Al samples made of the solution hardened Al alloy, AA5754, were first rough electropolished in air at room temperature prior to transferring to the pre-cooled cryogenic FIB/SEM stage. The Al samples had been processed by accumulative roll bonding (ARB), which was first introduced by Tsuji et al. [30]. The microstructure and mechanical properties of the alloy used in this work were published by Hausöl et al. [31]. After 8 ARB-passes, a grain size in the normal rolling direction of 50–100 nm is achieved [31]. The small grain size was chosen to increase the likelihood that a grain boundary was present within the final atom probe tip. The sample was sharpened using a Ga-ion FIB/SEM at -140˚C. Rather high energy ion beam parameters of 30 kV and 300pA were used during the final ion milling to cause the largest amount of Ga-implantation, which is usually undesirable [10] when preparing atom probe samples and often avoided by using lower voltages [32, 33]. However, knock on damage by the Ga ions was not a concern, since the main objective was to introduce a large amount of Ga ions as a "worst case scenario" for liquid metal embrittlement.

## 3. Results and discussion

After preparing the sharpened atom probe sample in the FIB at -140˚C (133 K), the cryogenic- and environmental-transfer system was used to transfer the sample to the LEAP while maintaining a measured temperature of -160˚C in the ACTD. For comparison, we have also carried out experiments where we have cryo-FIB milled specimens at -140˚C and transferred them into the atom probe at room temperature. These samples yielded only very small datasets without a grain boundary, or no data, as opposed to the cryo-transferred samples, which yielded good data in all cases. We attribute this to fractures caused by the well-known effect of Ga embrittling the grain boundaries, as an atom probe experiment puts a large mechanical stress onto the sample. The temperature above which this embrittlement effect comes into play has not yet been determined.

The results from an experiment with cryo-milling and cryo-transfer are presented in Fig 6. In Fig 6A, the microstructure of the nanocrystalline-Al is shown with the analysis direction of the APT reconstruction indicated. In this figure, a camera image of the sample in the analysis chamber of the atom probe clearly demonstrates that the sample was transferred

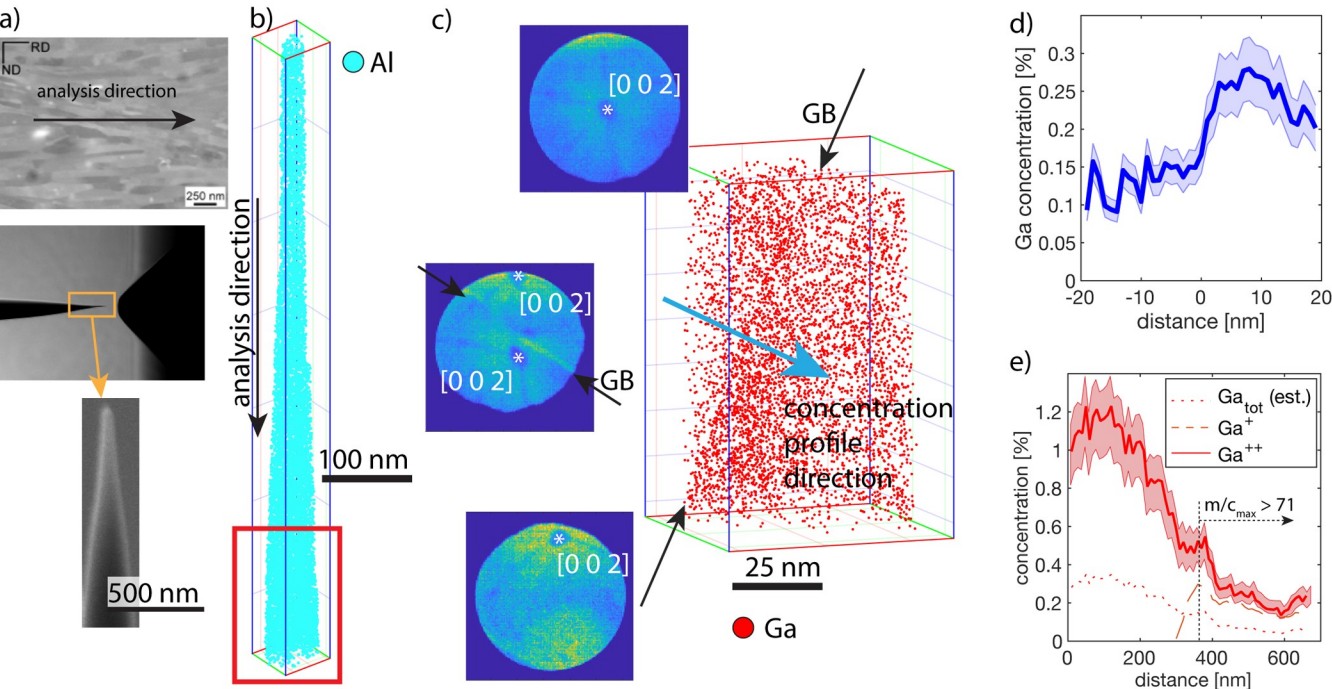

**Fig 6. Results of cryo-preparation of nanocrystalline-Al.** a) microstructure and resulting sample in the LEAP, showing no frost formation. b) overview of the reconstructed dataset. The red box corresponds to the data shown in c) detail of grain boundary showing change in crystallography and distribution of Ga. No segregation of Ga is visible at the grain boundary. d) Concentration profile of Ga across the grain boundary depicted in c). e) Concentration of Ga along the entire dataset, exhibiting a typical implantation profileNo enrichment at the grain boundary is observed.

without any discernible frost formation. The small volume design and in particular the copper-sheath surrounding the sample, effectively mitigate frost formation with this transfer system. Water molecules were also not encountered in the mass spectrum (see S5 and S6 Figs in S1 File). This is interesting as $H_2O$ is often found in APT data as a contaminant after conventional electrochemical or FIB preparation. A SEM image of the FIB milled sample is also shown, demonstrating the thin and highly elongated sample shape that is the result of the high primary $Ga^+$ ion energy (30keV) beam used to mill the sample. This shape is also evident in the shape of the 3D atom map of 100k randomly picked Al atoms in the dataset, presented in Fig 6B. The dataset was collected using voltage pulsing at a pulse frequency of 200kHz, a temperature of 40K, a detection rate of 0.5%, and a pulse fraction of 20%. From this 3D atom map, we have chosen a section containing a grain boundary for detailed analysis (Fig 6C). The presence of a grain boundary can clearly be observed through the change of the location of the crystallographic poles visible in the field desorption patterns [34]. Three field desorption patterns corresponding to locations above, in the middle and below the grain boundary are displayed together with the 3D distribution of the all detected Ga atoms in Fig 6C. A movie of the evolution of the field desorption patterns during the experiment and a movie of the 3D distribution of Ga as well as the APT data are included in the S1 File. In the 3D Ga distribution, there is already some structure visible, indicating the presence of a grain boundary. Most importantly, it is clear that Ga has not segregated at the grain boundary, in stark contrast to previous TEM [21, 22] and APT [23–25] investigations of aluminum alloys prepared with a Ga-FIB at room temperature. To quantify this claim, we have created a 1D concentration profile through the grain boundary as indicated in Fig 6C, shown in Fig 6D. This concentration profile confirms that no segregation is present at the

grain boundary, but a rise in Ga beyond the boundary along the direction of the ion penetration is observed. This is typical for ion implantation into a polycrystal, if the second crystal has an orientation towards the ion path with higher stopping power as compared to the first one. In fact, for the entire sample, a concentration profile typical for ion implantation is observed (Fig 6E). This concentration profile was taken along the analysis direction of the atom probe sample. Since Al is a relatively light element with low stopping power, the maximum concentration of the implantation is found at around 100 nm for 30keV $Ga^+$ ions. We do expect the implantation profile to significantly deviate from that of a flat sample though, since forward sputtering (escape of energetic ions through the side of the sample) would play a significant role in such a slim sample. In this sample, the overall implantation profile ($Ga_{tot}$) had to be estimated from the observed $Ga^{++}$ implantation profile, by assuming a constant charge state ratio between $Ga^+$ and $Ga^{++}$. This is owing to the fact that for the pulse frequency of 200kHz used in the experiment, the flight times of the $Ga^+$ ions were too long to be registered for about the first half of the data collected.

The cryogenic-temperatures during FIB-milling and transfer therefore sufficiently inhibit Ga diffusion to the grain boundaries, which is in-line with the recent results of Lilensten et al. [35]. Despite using a significantly higher final FIB voltage (i.e. implanting Ga deeper in the sample) than was used in the work of Lilensten, Ga diffusion to the grain boundary was still inhibited in part due to the cryogenic transfer of the specimens to the LEAP in this work. Taken together, these results have tremendous implications for the study of Al alloys, for example in the research areas of intergranular corrosion and strengthening mechanisms, which are of major importance in the automotive industry [29]. The ability to accurately characterize grain boundaries in Al alloys, will undoubtedly enable the elucidation of intergranular corrosion mechanisms. While additional investigations are required to identify the exact influence of FIB-milling and transfer temperatures on Ga diffusivity in Al alloys, these preliminary results show that maintaining cryogenic temperatures is a way to obtain Ga-free grain boundaries in such materials without resorting to alternative plasma-based FIB systems [36–38].

## 4. Conclusions and outlook

In this study, a custom-built versatile transfer system that enables quick transfer of samples that are sensitive to air or thermal exposure between sample preparation stations such as a Zeiss Crossbeam FIB/SEM and a local electrode atom probe (LEAP 4000X HR) was presented. The transfer system can easily be adapted to interface with any other instrumentation possessing a standard KF 16 port. In our lab, this includes a cryogenic electropolishing station, a field ion microscope and a coating system. A FIB transfer device (FIBTD) attaches directly to the existing FIB load lock, obviating the need for a specific cryo port on the microscope. Modifications to the LEAP enable quick, environmental transfer of samples directly into the buffer chamber using an actively cooled transfer device (ACTD) and a cryogenically cooled transfer stage in the LEAP. The ACTD maintains temperatures below -150˚ C throughout the transfer process. Using the transfer system, it was possible to observe Ga-free grain boundaries in an Al alloy, despite the samples being prepared with a Ga-based FIB.

Compared to other systems presented in literature, the ACTD is significantly smaller and incorporates close cryo-shielding of the sample to prevent frost formation on the sample during transfer. Additionally, the system can be easily modified to suit various experimental or analysis conditions and sample geometries since it is based on standard or easily milled components. This flexibility will be key as we study new materials systems such as those that are normally liquid at room temperature.

## Supporting information

**S1 File.**
(DOCX)

**S1 Video.**
(MOV)

**S2 Video.**
(MOV)

**S3 Video.**
(MOV)

## Author Contributions

**Conceptualization:** Chandra Macauley, Peter Felfer.

**Data curation:** Chandra Macauley.

**Formal analysis:** Chandra Macauley, Peter Felfer.

**Funding acquisition:** Peter Felfer.

**Investigation:** Chandra Macauley, Martina Heller, Alexander Rausch, Frank Kümmel.

**Methodology:** Chandra Macauley, Martina Heller, Alexander Rausch, Peter Felfer.

**Project administration:** Chandra Macauley.

**Resources:** Frank Kümmel.

**Software:** Martina Heller, Alexander Rausch, Peter Felfer.

**Supervision:** Peter Felfer.

**Validation:** Alexander Rausch.

**Visualization:** Chandra Macauley, Martina Heller, Alexander Rausch, Peter Felfer.

**Writing – original draft:** Chandra Macauley.

**Writing – review & editing:** Martina Heller, Frank Kümmel, Peter Felfer.

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
