## [Decision Letter · Decision Letter 0]

9 Nov 2020

PONE-D-20-27616

A versatile cryo-transfer system, connecting cryogenic focused ion beam sample preparation to atom probe microscopy

PLOS ONE

Dear Dr. Macauley,

Thank you for submitting your manuscript to PLOS ONE. After careful consideration, we feel that it has merit but does not fully meet PLOS ONE’s publication criteria as it currently stands. Therefore, we invite you to submit a revised version of the manuscript that addresses the points raised during the review process.

We look forward to receiving your revised manuscript.

Kind regards,

Leigh T. Stephenson

Academic Editor

PLOS ONE

Reviewers' comments:

Reviewer's Responses to Questions

**Comments to the Author**

1. Is the manuscript technically sound, and do the data support the conclusions?

Reviewer #1: Yes

Reviewer #2: Partly

2. Has the statistical analysis been performed appropriately and rigorously? 

Reviewer #1: N/A

Reviewer #2: N/A

3. Have the authors made all data underlying the findings in their manuscript fully available?

Reviewer #1: Yes

Reviewer #2: Yes

4. Is the manuscript presented in an intelligible fashion and written in standard English?

Reviewer #1: Yes

Reviewer #2: Yes

5. Review Comments to the Author

Reviewer #1: The manuscript titled “A versatile cryo-transfer system, connecting cryogenic focused ion beam sample preparation to atom probe microscopy” has been reviewed. The manuscript describes the development of unique hardware that enables the protected transfer of specimens between a cryogenic FIB/SEM and a LEAP for atom probe (AP) analysis. Specifically, specimen transfer to either the FIB/SEM or AP is performed using separate custom specimen shuttle suitcase devices (called “transfer arms” in the manuscript). The transfer device for FIB/SEM transfer maintains the specimen at passive cryogenic temperatures and passively at high vacuum conditions. The transfer device for the AP is capable of maintaining a specimen actively at cryogenic temperatures (-160 °C), while passively holding high vacuum conditions. Details of how the two different transfer devices are docked to the FIB/SEM and AP instruments are also discussed. In addition, a unique custom specimen carrier (called a double nipple) was developed to allow transfer between the FIB/SEM transfer device and the AP transfer device. For the latter, the design of the transfer device enables the specimen to be fully isolated within an actively cooled sub-volume so as to eliminate frost contamination during transfer. The efficacy of these devices, chambers, and described protocols are demonstrated via the analysis of an Al alloy intentionally exposed to relatively high Ga ion flux while at cryogenic temperatures, to show that Ga does not segregate to GBs. Appropriately referenced past reports by others show that Ga does indeed segregate to GBs in Al, when exposed to high Ga ion flux at room temperature.

Overall, the description of all the custom hardware and development of a unique approach is impressive. Despite other groups reporting similar developments for the FIB-based cryogenic preparation and environmentally protected transfer of specimens for APT analysis, the approach by the authors is unique and offers advantages (e.g. specimen shielding and small foot print) and is worthy of publication.

However, before this manuscript should be considered for publication, I would argue that the authors address the following comments and concerns outlined below. The order in which these concerns are described follow the chronological order encounter in the manuscript.

1. Sentence beginning with “Preparation of non-conductive or site-specific APT samples…” is awkward and makes too broad of assumption. As written, it implies that one cannot make site-specific samples from non-conductive, conductive, or semiconductive specimens for that matter. Additionally, the use of “exclusively” when referring to use of a FIB/SEM as being most easily or even exclusivity achieved is too broad. I agree that for site specific analysis, this is the true, but there are other material specimens worthy of APT analysis, such as pure metals, which do not require site-specific targeting, where electropolishing is much easier.

2. References 7-10: These references should be placed at the end of the sentence since they all apply to all subjects in this sentence.

3. Regarding sentence: “Some cryogenic- and environmental-transfer systems are commercially available and rely on a shuttle or suitcase that docks with various instruments and maintains a user-defined environment during transfer [11–14].” This is a good opportunity to help perpetuate a correct (logical) description with explicit nomenclature. I would argue that "Shuttle" refers to something that is used to secure a specimen (e.g. APT specimen pucks, can be called APT specimen shuttle pucks, where the word "puck" really describes a generic shape (disc-shaped), where "shuttle" describes the action (to transport). Suitcase describes a device which could hold other objects within it (i.e. isolated from the environment). So I would strongly suggest using the phrase "specimen shuttle suitcase device" to describe it. My understanding is that other publications describe such a device using various phrases, but if you were to use this phrase, maybe the community can adopt it in an effort to somewhat standardize terminology. This concept will be reiterated below.

4. Regarding the paragraph and discussion within that starts with “Some cryogenic- and environmental-transfer systems are…”: Since you go into superficially describing some of these systems in detail, please provide a more detailed comparison of the various transfer systems as suitcase devices that are: a) completely passive (e.g quorum; Perea et al.); b) actively cooled only (Leica, Gerstl et al.); c) actively cooled and pumped (Ferrovac; Stephenson et al.). Additionally, describing pros and cons will provide the context and clear distinction of the approach described here.

5. Typo: the word “Northwestern” is wrong. It should be Northwest, so that it reads “Pacific Northwest National Laboratory of the US Department of Energy…”

6. Regarding the sentence starting with “In this paper, we describe a custom-built, robust, and versatile transfer system…”, the word “robust” as used here doesn’t seem relevant as a descriptor. Please provide evidence of the system being ‘robust’.

7. Regarding sentence starting with “To demonstrate the cryogenic-, high vacuum-transfer capabilities of the system,”: It would be good to define "cryogenic" in this context. I suspect you mean that your device has active cooling. This would be more precise, otherwise given your implied definition here, even the Quorum transfer system is 'cryogenic' even though it is NOT actively cooled. This is an opportunity to highlight positive differences your system brings.

8. Regarding paragraph starting with “The ability to quickly make such environmentally-controlled transfers between instruments enables the application of atom probe tomography to previously inaccessible but increasingly important research fields beyond materials science, such as chemistry and biology.” While this is true, several other groups have now shown this. BUT what is it about your contribution here that is different? How does it improve or better enable new science? Again, use this as an opportunity to highlight positive differences your system brings.

9. Regarding the sentence beginning with “The cold-chain consists of optional initial cooling of the sample to the desired temperature, e.g. by plunge freezing.” It is confusing to use the phrase “cold-chain” as a descriptor of a protocol, without being explicit about what that means. Please consider rewriting this to say something like Here we use the phrase “cold-chain” to describe the protocol of XXXX. If this not the first instance of using this phrase as I describe here, then consider defining it in the location where it first appears. Additionally, other instances of this phrase are unhyphenated. Please be consistent.

10. Regarding the general organization: The organization of the manuscript is somewhat confusing, making it challenging for readers to understand the design and implementation of all the parts. To aid, please create separate sections describing: 1) modifications to the FIB/SEM; 2) modification to the APT system; 3) design of ancillary equipment such as sputter coater and electropolishing system; 4)design and utilization of the specimen shuttle suitcase device for transfer of specimens between the various tools described above.

11. Regarding sentence “Custom, pre-tilted (54°, which is the angle between the electron and ion beams) and non-pre-tilted cryo-shuttles with threads to accommodate the specimens were made to allow annular ion beam milling perpendicular to the sample surface without tilting the stage.” Please provide a picture description of this as a separate figure for both types of specimen shuttles; consider amending such images/drawings in existing figures. Additionally, nomenclature is confusing when describing 'shuttles'. This was brought up already above.

12. General comments: You describe your transfer devices as “transfer arms”. The phrase transfer arms, I believe, sells the design and function short. What I mean is that what you describe is more of a device, in that it serves much more purpose than just an arm that transfers specimens. This again, this is where you have the opportunity to be more precise and provide descriptive nomenclature. Strongly consider using the phrase “specimen shuttle suitcase transfer device” to define your “transfer arms”, and you can then shorten this description as “transfer device”.

13. Regarding the sentence “The generated vacuum of ~ 10-4 mbar thereby has proven to be sufficient to prevent frost built up on the specimen.” Please provide a statement that evidence of this is shown below during the exemplary analysis of an Al alloy. Otherwise, the reader is left wondering why such a conjectured statement was made.

14. Be explicit in referring to which “transfer arm” you are referring to in the sentence “In this first transfer arm, no active cooling or pumping is available”.

15. Regarding the sentence “Measured heating rates of the shuttle while uncooled at the end of the PEEK end-effector at a vacuum of around 10-2 mbar are in the range of 2.5°C/min.” Considering conductive thermal heat transfer, the rate in a change of temperature depends on the temperature of the specimen relative to the temperature of the object it is in contact with. This change in temperature per unit time is not linear as you imply. Also, please revise this sentence to be more clear. I am to understand that your specimen is starting off as cold, so it has been 'cooled'; it is not 'uncooled' as you say. Be more precise by saying your specimen "is not actively cooled".

16. Awkward sentence structure. Please revise: “Switzerland). While these valves have the same leak tightness as their metal flanged, fluoroelastomer (FKM)-sealed counterparts, they cannot be heated beyond 80°C, limiting possible bake-outs. However, moving samples from the FIB/SEM with ultimate pressures not below 10-7 mbar, this is not of any concern.

17. Regarding Fig. 3. Please provide an inset image that is a zoomed in region of where the double nipple screws into your transfer device, and how it is enclosed (shielded). It is hard to decern without any arrows or labels in the figure. This will help the reader to make more sense of the respective description in the paragraphs that follow.

18. Regarding the sentences “The sample is held in a double-nipple, Fig 3C, that has a 5 mm diameter right-handed screw thread on one side, and a left-handed screw thread on the other. Such a design enables the double nipple to be transferred between the copper end-component shown in Fig 3A, and FIB or LEAP sample holders with one continuous twisting motion.” Please provide a reference in the figures to how this is done. Consider adding additional figure panels to help explain this visually.

19. Regarding the sentence “This will hold true so long as sufficient vacuum is maintained that leads to molecular flow of the residual gas atoms.” Wouldn't this be the case until the pressure is well above atm pressure (i.e. turbulent or laminar flow)? Maybe I’m mistaken, but as such, this statement does not make sense as used here.

20. Regarding the sentence that ends with “…the roughing pump maintains the vacuum.” What explicitly are you referring to?...FIB/SEM load lock?

21. Regarding the sentence: “During transfer, the FIB cryo-shuttle is rotated 90��along the axis of the FIB transfer arm, so that the pre-tilted double nipple and atom probe sample are parallel to the axis of the LEAP transfer arm, Fig 4C. The threaded copper sheath of the LEAP transfer”. Please add labels (with arrows) that indicate the orientation and direction of the LEAP transfer arm, the SEM transfer arm, and the location of the SEM load lock/SEM chamber.

22. Regarding the sentence: “The sample transfer takes place in the buffer chamber,…”. Please explain the what the “buffer chamber is. This phrase is jargon for the specific instrument and is familiar to the APT community explicitly as you use it. Consider stating, "The sample transfer takes place in what is commonly referred to as the buffer chamber of the Local Electrode Atom Probe,"

23. Regarding the sentence: “This is the fastest conceivable route.” This is not explicitly true. Direct transfer into the analysis chamber would be the quickest route, but is not practical given LEAP design. Also, this route would create potential contamination of the AC. For these reasons, BC connection makes the most sense, as going through the LL would create extra steps of having to use the LEAP-specific specimen puck carousels. .

24. Regarding the sentence: “The stage can be cooled with liquid nitrogen via a vessel (4 in Fig 5) that is also connected to the linear drive of the stage.” Please show schematic details of this to provide details that any reader would as about how the stage is "moved".

25. Regarding the sentence: “During the hand-over, the LEAP puck is held in place…”. Please provide additional details describing how this done. Is it done while holding the stage on the LEAP transfer arm? or does the LEAP transfer arm then grab the sample from the "movable" stage after hand off, from which it can then be loaded into the AC? Again, showing schematically is necessary for clarification.

26. Regarding the sentence: “To demonstrate the efficacy of the cryogenic-, environmental-transfer capabilities of the system, we chose to show the prevention of liquid metal embrittlement in the Al – Ga system.”. I’m having trouble understanding how APT analysis shows “the prevention of liquid metal embrittlement”. Such a determination would require some mechanical testing analysis to show this. Instead APT is able to confirm the distribution of Ga relative to GBs, and an inference would then be made that the lack of GB enrichment of Ga leads to less mechanical embrittlement. Maybe you are referring to the observations/statements implying a relatively high APT analysis yield compared to other studies which showed low yield at relatively low applied biases (i.e. fractured at lower stresses). Please be explicit in explaining such phenomena.

27. General statements regarding Figures:

a. Fig. 1b. label the FIB/SEM transfer device connected to the LL of the FIB/SEM

b. Provide image of the 90deg type of FIB/SEM specimen pucks; Figure 2 only shows the 54 degree version

c. Add error bars (consider shaded line bands) to the plots in Fig. 6D and E.

Reviewer #2: 1. There is some analysis regarding the distribution of Ga++ and the presence of MgOH complex ions within the sample that I believe needs to be conducted again in relation to the suggestions I have made in the uploaded comments.

4. Although well written, it is difficult to follow the train of thought due to the use of multiple and overlapping names for key components. This needs to be fixed prior to publication.

6. PLOS authors have the option to publish the peer review history of their article (what does this mean?). If published, this will include your full peer review and any attached files.

Reviewer #1: **Yes: **Daniel Perea

Reviewer #2: **Yes: **Ingrid E. McCarroll

---

## [Author Response · Author response to Decision Letter 0]

24 Dec 2020

Reviewer 1: 

The manuscript titled “A versatile cryo-transfer system, connecting cryogenic focused ion beam sample preparation to atom probe microscopy” has been reviewed. The manuscript describes the development of unique hardware that enables the protected transfer of specimens between a cryogenic FIB/SEM and a LEAP for atom probe (AP) analysis. Specifically, specimen transfer to either the FIB/SEM or AP is performed using separate custom specimen shuttle suitcase devices (called “transfer arms” in the manuscript). The transfer device for FIB/SEM transfer maintains the specimen at passive cryogenic temperatures and passively at high vacuum conditions. The transfer device for the AP is capable of maintaining a specimen actively at cryogenic temperatures (-160 °C), while passively holding high vacuum conditions. Details of how the two different transfer devices are docked to the FIB/SEM and AP instruments are also discussed. In addition, a unique custom specimen carrier (called a double nipple) was developed to allow transfer between the FIB/SEM transfer device and the AP transfer device. For the latter, the design of the transfer device enables the specimen to be fully isolated within an actively cooled sub-volume so as to eliminate frost contamination during transfer. The efficacy of these devices, chambers, and described protocols are demonstrated via the analysis of an Al alloy intentionally exposed to relatively high Ga ion flux while at cryogenic temperatures, to show that Ga does not segregate to GBs. Appropriately referenced past reports by others show that Ga does indeed segregate to GBs in Al, when exposed to high Ga ion flux at room temperature.

Overall, the description of all the custom hardware and development of a unique approach is impressive. Despite other groups reporting similar developments for the FIB-based cryogenic preparation and environmentally protected transfer of specimens for APT analysis, the approach by the authors is unique and offers advantages (e.g. specimen shielding and small foot print) and is worthy of publication.

However, before this manuscript should be considered for publication, I would argue that the authors address the following comments and concerns outlined below. The order in which these concerns are described follow the chronological order encounter in the manuscript.

1. Sentence beginning with “Preparation of non-conductive or site-specific APT samples…” is awkward and makes too broad of assumption. As written, it implies that one cannot make site-specific samples from non-conductive, conductive, or semiconductive specimens for that matter. Additionally, the use of “exclusively” when referring to use of a FIB/SEM as being most easily or even exclusivity achieved is too broad. I agree that for site specific analysis, this is the true, but there are other material specimens worthy of APT analysis, such as pure metals, which do not require site-specific targeting, where electropolishing is much easier.

We have modified the sentence to make it clear that electropolishing is used for conductive, non-site specific samples but that the FIB/SEM is often used to make APT samples from non-conductive or site-specific samples. 

2. References 7-10: These references should be placed at the end of the sentence since they all apply to all subjects in this sentence. 

Fixed. 

3. Regarding sentence: “Some cryogenic- and environmental-transfer systems are commercially available and rely on a shuttle or suitcase that docks with various instruments and maintains a user-defined environment during transfer [11–14].” This is a good opportunity to help perpetuate a correct (logical) description with explicit nomenclature. I would argue that "Shuttle" refers to something that is used to secure a specimen (e.g. APT specimen pucks, can be called APT specimen shuttle pucks, where the word "puck" really describes a generic shape (disc-shaped), where "shuttle" describes the action (to transport). Suitcase describes a device which could hold other objects within it (i.e. isolated from the environment). So I would strongly suggest using the phrase "specimen shuttle suitcase device" to describe it. My understanding is that other publications describe such a device using various phrases, but if you were to use this phrase, maybe the community can adopt it in an effort to somewhat standardize terminology. This concept will be reiterated below.

The terminology used in the paper was indeed confusing and we agree that more standardized terminology would benefit the community. We have made changes so that the term “shuttle” is now only used when describing components in which specimens are secured (eg. FIB cryo-shuttle, APT specimen shuttle pucks). Any device used to transport specimen shuttles between instruments is now referred to as a “specimen shuttle transfer device” or “transfer device” for short. To differentiate between the two transfer devices, they are now referred to as the ‘FIB transfer device’ and the ‘actively cooled transfer device (ACTD)’ throughout the paper. 

4. Regarding the paragraph and discussion within that starts with “Some cryogenic- and environmental-transfer systems are…”: Since you go into superficially describing some of these systems in detail, please provide a more detailed comparison of the various transfer systems as suitcase devices that are: a) completely passive (e.g quorum; Perea et al.); b) actively cooled only (Leica, Gerstl et al.); c) actively cooled and pumped (Ferrovac; Stephenson et al.). Additionally, describing pros and cons will provide the context and clear distinction of the approach described here.

5. Typo: the word “Northwestern” is wrong. It should be Northwest, so that it reads “Pacific Northwest National Laboratory of the US Department of Energy…”

Apologies for the typo. It is corrected. 

6. Regarding the sentence starting with “In this paper, we describe a custom-built, robust, and versatile transfer system…”, the word “robust” as used here doesn’t seem relevant as a descriptor. Please provide evidence of the system being ‘robust’. The word robust was removed.

7. Regarding sentence starting with “To demonstrate the cryogenic-, high vacuum-transfer capabilities of the system,”: It would be good to define "cryogenic" in this context. I suspect you mean that your device has active cooling. This would be more precise, otherwise given your implied definition here, even the Quorum transfer system is 'cryogenic' even though it is NOT actively cooled. This is an opportunity to highlight positive differences your system brings. “actively cooled” was added to highlight the difference between our system and other passive systems. 

8. Regarding paragraph starting with “The ability to quickly make such environmentally-controlled transfers between instruments enables the application of atom probe tomography to previously inaccessible but increasingly important research fields beyond materials science, such as chemistry and biology.” While this is true, several other groups have now shown this. BUT what is it about your contribution here that is different? How does it improve or better enable new science? Again, use this as an opportunity to highlight positive differences your system brings.

The sentence was removed and replaced with the following to highlight the advantages of our system: 

While the ability to quickly make environmentally controlled transfers between instruments has been shown by other groups, our system has unique advantages such as a small footprint and close shielding of the sample that prevents frost formation. Since it uses standard parts or components that can easily be milled, the system is highly adaptable to new experimental arrangements. Additionally, the system does not require a dedicated cryo port on the FIB/SEM, as it interlocks directly with the existing load lock door, a feature that is ideal for multi-user instruments that are not exclusively used for cryo experiments.

9. Regarding the sentence beginning with “The cold-chain consists of optional initial cooling of the sample to the desired temperature, e.g. by plunge freezing.” It is confusing to use the phrase “cold-chain” as a descriptor of a protocol, without being explicit about what that means. Please consider rewriting this to say something like Here we use the phrase “cold-chain” to describe the protocol of XXXX. If this not the first instance of using this phrase as I describe here, then consider defining it in the location where it first appears. Additionally, other instances of this phrase are unhyphenated. Please be consistent. A sentence defining “cold chain” was added and the hyphen removed for consistency. 

10. Regarding the general organization: The organization of the manuscript is somewhat confusing, making it challenging for readers to understand the design and implementation of all the parts. To aid, please create separate sections describing: 1) modifications to the FIB/SEM; 2) modification to the APT system; 3) design of ancillary equipment such as sputter coater and electropolishing system; 4)design and utilization of the specimen shuttle suitcase device for transfer of specimens between the various tools described above. The current organization of the paper follows the sample through the FIB manufacture and subsequent transfer to the LEAP. The section titles of the methods section have been changed to: 

The following sentence was added to the introduction: “The paper is organized in the order of components/modifications used during a cryo-FIB preparation of an atom probe sample and subsequent transfer into the LEAP.”

11. Regarding sentence “Custom, pre-tilted (54°, which is the angle between the electron and ion beams) and non-pre-tilted cryo-shuttles with threads to accommodate the specimens were made to allow annular ion beam milling perpendicular to the sample surface without tilting the stage.” Please provide a picture description of this as a separate figure for both types of specimen shuttles; consider amending such images/drawings in existing figures. Additionally, nomenclature is confusing when describing 'shuttles'. This was brought up already above.

The figure has been modified accordingly. 

12. General comments: You describe your transfer devices as “transfer arms”. The phrase transfer arms, I believe, sells the design and function short. What I mean is that what you describe is more of a device, in that it serves much more purpose than just an arm that transfers specimens. This again, this is where you have the opportunity to be more precise and provide descriptive nomenclature. Strongly consider using the phrase “specimen shuttle suitcase transfer device” to define your “transfer arms”, and you can then shorten this description as “transfer device”.

See response to point 3). 

13. Regarding the sentence “The generated vacuum of ~ 10-4 mbar thereby has proven to be sufficient to prevent frost built up on the specimen.” Please provide a statement that evidence of this is shown below during the exemplary analysis of an Al alloy. Otherwise, the reader is left wondering why such a conjectured statement was made. 

The following was added to the end of the aforementioned sentence: “…, as shown below in the analysis of the Al alloy.”

14. Be explicit in referring to which “transfer arm” you are referring to in the sentence “In this first transfer arm, no active cooling or pumping is available”.

The sentence was changed as follows: “In this FIB transfer device, no active cooling or pumping is available.” Additionally the second transfer arm is now only referred to as the ‘actively cooled transfer device, ACTD’, as clarified in point 3). 

15. Regarding the sentence “Measured heating rates of the shuttle while uncooled at the end of the PEEK end-effector at a vacuum of around 10-2 mbar are in the range of 2.5°C/min.” Considering conductive thermal heat transfer, the rate in a change of temperature depends on the temperature of the specimen relative to the temperature of the object it is in contact with. This change in temperature per unit time is not linear as you imply. Also, please revise this sentence to be more clear. I am to understand that your specimen is starting off as cold, so it has been 'cooled'; it is not 'uncooled' as you say. Be more precise by saying your specimen "is not actively cooled". The sentences have been revised to be more specific. 

“To measure the efficacy of this thermal isolation, specimen temperature was measured while a FIB cryo shuttle, pre-cooled by the cryo stage to -118°C, warmed up (since it was not actively cooled) during a simulated hand-off to the ACTD. As shown in Fig S2, the pre-cooled shuttle on the PEEK end-effector of the FIBTD warms up at 2.3°C/min at a pressure below 10-5 mbar. “ 

16. Awkward sentence structure. Please revise: “Switzerland). While these valves have the same leak tightness as their metal flanged, fluoroelastomer (FKM)-sealed counterparts, they cannot be heated beyond 80°C, limiting possible bake-outs. However, moving samples from the FIB/SEM with ultimate pressures not below 10-7 mbar, this is not of any concern.

The sentences have been revised and are now: ‘Although these KF valves have the same leak tightness as their CF metal flanged, fluoroelastomer (FKM)-sealed counterparts, they cannot be heated beyond 80°C, limiting possible bake-outs. Such higher temperature bake-outs are only needed to reach ultra high vacuum (<10-9 mbar) conditions. Since the ACTD is being connected to the FIB/SEM, which has an ultimate pressure of 10-7 mbar, such bake-outs are not required.’ 

17. Regarding Fig. 3. Please provide an inset image that is a zoomed in region of where the double nipple screws into your transfer device, and how it is enclosed (shielded). It is hard to decern without any arrows or labels in the figure. This will help the reader to make more sense of the respective description in the paragraphs that follow. 

An inset is now shown in Fig. 3C including labels and arrows. 

18. Regarding the sentences “The sample is held in a double-nipple, Fig 3C, that has a 5 mm diameter right-handed screw thread on one side, and a left-handed screw thread on the other. Such a design enables the double nipple to be transferred between the copper end-component shown in Fig 3A, and FIB or LEAP sample holders with one continuous twisting motion.” Please provide a reference in the figures to how this is done. Consider adding additional figure panels to help explain this visually.

While we have added annotations and additional panels to Fig 3 and Fig 4, we think the best way to understand the mechanism of sample hand-off is with a movie. This movie is now provided in the supplementary information for the paper. 

19. Regarding the sentence “This will hold true so long as sufficient vacuum is maintained that leads to molecular flow of the residual gas atoms.” Wouldn't this be the case until the pressure is well above atm pressure (i.e. turbulent or laminar flow)? Maybe I’m mistaken, but as such, this statement does not make sense as used here.

The sentence in question was removed. 

20. Regarding the sentence that ends with “…the roughing pump maintains the vacuum.” What explicitly are you referring to?...FIB/SEM load lock?

“…FIB/SEM load lock…” was added to clarify which pump provides the vacuum when moving samples INTO the FIB/SEM. 

21. Regarding the sentence: “During transfer, the FIB cryo-shuttle is rotated 90��along the axis of the FIB transfer arm, so that the pre-tilted double nipple and atom probe sample are parallel to the axis of the LEAP transfer arm, Fig 4C. The threaded copper sheath of the LEAP transfer”. Please add labels (with arrows) that indicate the orientation and direction of the LEAP transfer arm, the SEM transfer arm, and the location of the SEM load lock/SEM chamber.

Additional figure panels, labels, and arrows have been added to Fig 4 to better explain the orientation.

22. Regarding the sentence: “The sample transfer takes place in the buffer chamber,…”. Please explain the what the “buffer chamber is. This phrase is jargon for the specific instrument and is familiar to the APT community explicitly as you use it. Consider stating, "The sample transfer takes place in what is commonly referred to as the buffer chamber of the Local Electrode Atom Probe,"

The sentence has been changed to: “The sample transfer takes place in what is commonly referred to as the buffer chamber of the LEAP, after which…”. The LEAP acronym was defined in the introduction and methods section of the paper. 

23. Regarding the sentence: “This is the fastest conceivable route.” This is not explicitly true. Direct transfer into the analysis chamber would be the quickest route, but is not practical given LEAP design. Also, this route would create potential contamination of the AC. For these reasons, BC connection makes the most sense, as going through the LL would create extra steps of having to use the LEAP-specific specimen puck carousels. .

We agree with the reviewer and have deleted “this is the fastest conceivable route”. 

24. Regarding the sentence: “The stage can be cooled with liquid nitrogen via a vessel (4 in Fig 5) that is also connected to the linear drive of the stage.” Please show schematic details of this to provide details that any reader would as about how the stage is "moved". Additional figure panels have been added to illustrate the stage movement.

25. Regarding the sentence: “During the hand-over, the LEAP puck is held in place…”. Please provide additional details describing how this done. Is it done while holding the stage on the LEAP transfer arm? or does the LEAP transfer arm then grab the sample from the "movable" stage after hand off, from which it can then be loaded into the AC? Again, showing schematically is necessary for clarification. Additional details have been provided both in the text and figure. A movie of the transfer process has been included in the supplementary materials. 

26. Regarding the sentence: “To demonstrate the efficacy of the cryogenic-, environmental-transfer capabilities of the system, we chose to show the prevention of liquid metal embrittlement in the Al – Ga system.”. I’m having trouble understanding how APT analysis shows “the prevention of liquid metal embrittlement”. Such a determination would require some mechanical testing analysis to show this. Instead APT is able to confirm the distribution of Ga relative to GBs, and an inference would then be made that the lack of GB enrichment of Ga leads to less mechanical embrittlement. Maybe you are referring to the observations/statements implying a relatively high APT analysis yield compared to other studies which showed low yield at relatively low applied biases (i.e. fractured at lower stresses). Please be explicit in explaining such phenomena.

The sentence has been changed to “…FIB-induced Ga segregation to the grain boundaries in the Al – Ga system.”. While it has been well documented that Ga-segregation causes embrittlement of Al alloys, the review is correct that we did not explicitly study the mechanical properties. It is true that our analysis yield was substantially better for cryo-prepared and transferred samples compared to samples either prepared with room T Ga-based FIB or transferred at room temperature. 

27. General statements regarding Figures:

a. Fig. 1b. label the FIB/SEM transfer device connected to the LL of the FIB/SEM Revised.

b. Provide image of the 90deg type of FIB/SEM specimen pucks; Figure 2 only shows the 54 degree version Both FIB cryo shuttles are now shown

c. Add error bars (consider shaded line bands) to the plots in Fig. 6D and E. Shaded line bands have been added. 

Reviewer 2 

Overall Statement

This study presents results of an in-house design of a unique cryogenic transfer system, primarily connecting a Ga-FIB/SEM to an atom probe. The work also showcases the elimination of Ga segregation to grain boundaries, commonly observed in room temperature ion beam milling and specimen transfer. Although the work overall is very promising and relevant to the scientific community, I do have some concerns related to the content. My primary concerns are: 1) there are some results that I believe need further analysis and clarification before they are ready for publication, 2) the overall presentation of the work is confusing and requires clarification, specifically in the area of naming conventions and overall workflow, and 3) the conclusions seem rather broad and do not directly reflect the findings in the paper.

Detailed Comments

1. The acronym TEM is not introduced in the document. This could be introduced in the 1st paragraph of the introduction.

The acronym has been included.

2. 2nd paragraph of the introduction it sounds as though MPIE has a commercial interest in the Ferrovac system: “In collaboration with CAMECA and the Max Planck Institute for Iron Research in Dusseldorf, Ferrovac offers a large…”. It might be better to say that in collaboration with these companies/institutes that “Ferrovac has developed a …”. (Unless of course MPIE has a commercial interest that I am not aware of…?) 

The suggested changes have been made.

3. The last sentence of the 2nd paragraph in the introduction is a bit vague: “Pacific Northwestern National Laboratory of the US Department of Energy that offers even more possibilities”. Please be more specific about the further possibilities offered by this system.

4. In the sentence ending “through a cryogenic shuttle device designed by the authors, which is introduced below”, remove “which is introduced below” as the first object introduced is the transfer arm, which may then be mistaken for the shuttle device.

The suggested changes have been made.

5. In the sentence “A general overview of the system including relevant temperatures and pressures at the various steps of sample preparation …”, please add ‘cryogenic/vacuum transfer’ before ‘system’.

The suggested changes have been made.

6. It is unclear how you would actually plunge freeze the sample prior to transfer to the FIB. Would this be accomplished in some kind of controlled environment?

As plunge freezing was not used in the specific experiments presented in this paper, we removed it from the paper for clarity. 

7. Fig. 1 is very difficult to comprehend and would benefit from rearranging the information. This image is critical to the reader’s capacity to understand the overall interconnectedness of the system and the overall workflow. Care should be taken to make sure it is as clear as possible. 

a. Fig. 1a should be removed and this information incorporated into a single workflow diagram. (Note that temperatures are already incorporated into Figures b,c, and d resulting in double-up of information)

b. I would suggest placing the FIB/SEM at the top of the image, with operating properties beside it. Separate the transfer arm from the FIB/SEM and place this below the FIB/SEM with the transfer shuttle beside it providing appropriate information for both. Place the LEAP at the bottom (‘CAMECA’ is not really necessary and just takes up space) with appropriate information. Join each component with arrows (as in current Fig. 1a, however make the arrows long enough that the required information can be provided adjacent to them. It would also be clearer if you separated the cryo LL from the LEAP for the purposes of clearly articulating the workflow and isolating the newly designed components from existing infrastructure.

c. Where you have written -196 dC - RT, please change this to -196 dC to RT, otherwise it looks like an equation

d. Where and how does plunge freezing fit into this workflow?

We appreciate the constructive feedback and have significantly changed the figure based on your suggestions. A workflow is shown in part a) of the figure, clearly indicating what components have been designed and built as part of this work. A schematic workflow is shown in part b), using the same color scheme, to indicate the temperatures and pressures at critical points. We have left the FIB transfer device connected to the FIB because one advantage of our system is that the FIB load lock does not need to be changed to accommodate the transfer device. As mentioned previously, plunge freezing has been removed from the diagram since it was not used in this study. 

8. In the sentence “After FIB milling and/or SEM characterization, the sample is moved out of the FIB/SEM using the same, thermally-insulated, but not actively-cooled transfer arm, in which the sample is handed off to another, actively-cooled transfer arm, with a standard KF16 vacuum flange (Fig 1C)”, the name transfer arm is used to describe two distinct parts of the system. Please be consistent with notation and refer to each component using a distinct naming convention. Alternatively, if you mean that the ‘sample is handed off to another, actively-cooled transfer arm that delivers the sample to the transfer shuttle’ then please be more specific in the detail.

The terminology used in the paper was indeed confusing. We have made changes so that the term “shuttle” is now only used when describing components in which specimens are secured (eg. FIB cryo-shuttle, APT specimen shuttle pucks). Any device used to transport specimen shuttles between instruments is now referred to as a “specimen shuttle transfer device” or “transfer device” for short. To differentiate between the two transfer devices, they are now referred to as the ‘FIB transfer device’ and the ‘actively cooled transfer device (ACTD)’ throughout the paper. 

9. In reference to point 8, a number of other items have different names in the text and in the image, please check through the text carefully and be consistent with all naming conventions throughout. The constant changing of names and similarity of names made following the thread of the article extremely difficult.

See response to previous point. 

10. Assuming that the cryo electropolishing station is the station used for initial plunge freezing (discussed in the main text), the image of the cryo electropolishing unit should be included in Figure 1.

As plunge freezing was not used in the described experiments, mention of it has been removed from the paper. 

11. Fig 2:

a. Add annotations to a)

b. Change port names to TS1 and TS2 to represent transfer shuttle 1 and 2. APT 1 and 2 suggest that it will be connected to the atom probe directly.

c. A better name could be chosen for the cryo-shuttle, at the moment it is too similar to transfer shuttle, which also operates at cryo temperatures.

Annotations have been added and APT 1 and 2 have been changed to ACTD1 and ACTD2, in agreement with the new name for this transfer device. See response to point 8. 

12. It is not clear how the transfer arm enables the analysis of multiple samples, as indicated in the following sentence “the transfer arm reduces the risk of chamber contamination and enables efficient characterization of multiple samples.” It appears to me that only one sample can be transferred at any one time.

The sentence has been changed to more clearly indicate why having a FIB transfer device is essential to analyzing multiple samples in a single FIB session. “The transfer device reduces the risk of chamber contamination and enables efficient characterization of multiple samples in a single FIB session, since without the means to transfer cold samples through the load-lock, the cold stage would need to be warmed to room temperature for sample exchanges and the entire analysis chamber vented.”

13. “This transfer arm can be used to plunge freeze liquid containing samples prior to characterization (see supplementary materials) and interfaces with the existing Zeiss 80 mm Airlock without further modification”. It is unclear to me how the transfer arm can be used to plunge freeze liquid containing samples without exposure to air. If air exposure is necessary in the process, then ice will build-up on the sample making the method redundant. Please explain further the initial experimental transfer step of samples into the transfer arm under a controlled environment.

See response to points 6 and 10 above.

14. “This pumping operation takes ca. 40s until a vacuum sufficient for transfer is established.” This is longer than the 30s, stated in the supplementary section, as the maximum time in order not to exceed the crystallization temperature of ice.

References to plunge freezing have been removed from the manuscript and supplementary information since it is not relevant for the experimental results shown in this paper. 

15. The supplementary section states the crystallization temperature of water ice to be -160 dC. Please provide a reference, as my understanding was that cryo EM scientists aim to keep the temperature below -150 dC, and that this is sufficient to maintain vitreous ice. Perhaps pressures should be considered in the statement of transition temperatures?

See previous response. Pressures are absolutely important. A new table showing the heating rate as a function of pressure is provided in the supplementary information. 

16. 2.2 Atom probe transfer shuttle. This shuttle can be attached to multiple instruments, perhaps a more generic name would be more appropriate.

We have changed the name to ‘actively cooled transfer device’.

17. Section 2.2: “So long the reservoir does not run dry, the sample is near liquid nitrogen temperature in the transfer shuttle.” The supplementary information indicates that the minimum temperature of the sample is approx. -170 dC, which is not so near liquid nitrogen temperature. Please add the specific temperature to the text. The specific temperature is now used. 

18. “However, moving samples from the FIB/SEM with ultimate pressures not below 10-7 mbar, this is not of any concern.” This sentence is confusing, please reword.

The sentence has been changed to: “Although these KF valves have the same leak tightness as their CF metal flanged, fluoroelastomer (FKM)-sealed counterparts, they cannot be heated beyond 80°C, limiting possible bake-outs. Such higher temperature bake-outs are only needed to reach ultra high vacuum (<10-9 mbar) conditions. Since the ACTD is being connected to the FIB/SEM, which has an ultimate pressure of 10-7 mbar, such bake-outs are not required.” 

19. “enabling temperatures of -160 °C during transfer.” This sentence is now in contrast to the statement, “So long the reservoir does not run dry, the sample is near liquid nitrogen temperature in the transfer shuttle.”. See response to point 16. 

20. The choice of colour for the tip in Fig 3b makes it difficult to see the tip against a similar background colour. Please change. 

A zoomed-in inset has been added, as well as labels, to make it easier to see the critical components. 

21. Fig. 3, consider indicating the location of the sample with an arrow and label for those not familiar with the appearance of APT samples.

Labels, arrows and additional panels have been added to Fig 3. 

22. In section 2.4 you refer to the existing atom probe transfer arm as “LEAP transfer arm”, this becomes particularly confusing now given that the transfer shuttle is often referred to as “LEAP transfer arm”. Please use distinct names. The name of the transfer device has now been changed to actively cooled transfer device (ACTD). 

23. “capable of in-situ heating and cryogenic cooling, and its electrical connections”, ‘and its electrical connections’ doesn’t make sense in this sentence. Please clarify. The sentence has been modified to emphasize that the additional flanges on the cryo transfer chamber include a viewport, a movable cryogenic/heating stage and the electrical connections required for the stage. 

24. “can be transferred to a pre-cooled, custom LEAP puck”, it would be better here if you said “can be screwed into a pre-cooled…” to remind readers that the sample is not transferred on a puck but via a double-nipple. The suggested change has been made. 

25. “During this hand-over, the LEAP puck is held in place by the actively-cooled cryogenic stage, to make the alignment of the double nipple and the puck easier.” It is not clear from this sentence what is making the alignment of the double nipple and puck easier. The comment about it making alignment easier was removed. Prior to the design and construction of the cryo stage, pucks were first pre-cooled in the AC and the handoff between the ACTD and the custom puck happened with the puck attached to the LEAP transfer rod. The puck moved slightly on the transfer rod, making the hand-off challenging. The cryo-stage therefore makes the hand-off easier by holding the puck in place.

26. “with the long axis of the sample indicated” Please indicate in this sentence that you mean the long axis of the atom probe tip. The sentence has been modified to include analysis direction instead of long axis. 

27. “This is interesting as H2O is often found in APT data as a contaminant after conventional electrochemical or FIB preparation”. Note that you have Mg2O peaks and what looks like a large Mg2H2O peak between the Ga2+ peaks and again MgOHx between 40-50 Da (double check). It is common in Mg containing data not to see H or O peaks, because these elements bond to and evaporate with the Mg in the sample. If you are inclined to continue to mention the lack of H2O in the dataset, you must also mention the presence of the Mg-O complex ions in the dataset.

Based on the following mass spectra, the peaks in question do not fit those of Mg2O, Mg2H2O or MgOHx. 

28. Please list the detection rate as a part of the atom probe parameters. Included. 

29. Figure 6:

a. This image is overly compact. Please separate the images or place c and d below a and b.

b. Make sure the scale bars do not overlap with the images.

c. Please redo the scale bar for c, I don’t believe this scale bar represents all three axes accurately.

The image has been modified slightly. The scale bar in c does accurately represent all three axes as it is an orthographic projection. 

30. Something odd is happening in the video of the desorption map. The edges of the map change significantly throughout the acquisition and poles seem to come and go in relation to the moving of the edges. A few things may cause this: 1) significant vibration of the tip, 2) significant fractures to the tip and 3) slow drifting of the tip. All of which will affect the data. Please address this in the text and discuss implications on the data.

The field desorption maps change because the custom cryo-shuttle pucks used in the LEAP experiments had not yet been optimized to fit perfectly. During the course of an atom probe experiment, the sample shuttle puck would move slightly due to being slightly too small for the puck holder in the LEAP. This issue has since been fixed with precisely sized custom LEAP shuttle pucks. 

31. I am concerned with the use of Ga++ as an indicator for Ga+, due to the overlap between Mg2O and Ga++. This analysis needs to be redone considering this overlap and its implications on Ga concentration. This is particularly of concern given that the peak for the heavier Ga isotope is greater than the peak of the lighter Ga isotope. Furthermore, Ga++ does not necessarily always evaporate after FIB preparation, so it could be that these peaks are entirely related to the Mg-O-H peaks.

See response to point 27. 

32. The second paragraph of the conclusion reads more like an introduction than a conclusion. I think that it is necessary to rewrite this paragraph to focus on the specific outcomes of this particular study, such as the observation of Ga free grain boundaries, the rapid transfer times, the shielding of the tip during transfer and the low temperatures maintained during transfer.

The second paragraph has been removed and replaced with a new paragraph that focuses specifically on the advantages of our system and the Ga-free grain boundary result. 

Supplementary section

1. The figures are not directly referenced in the text. This is particularly confusing when it is stated that “shown below are the temperature and pressure logs” when the below image is of the experimental set-up. Please be specific in referring to the images in the text.

Revised.

2. Again further names are given to the shuttle ‘atom probe shuttle’ and ‘LEAP transfer shuttle’, a consistent name is required for the transfer shuttle.

FIBTD, ACTD and FIB cryo shuttle are now used consistently throughout the paper and supplementary information.

---

## [Editor Report · Decision Letter 1]

4 Jan 2021

A versatile cryo-transfer system, connecting cryogenic focused ion beam sample preparation to atom probe microscopy

PONE-D-20-27616R1

Dear Dr. Macauley,

We’re pleased to inform you that your manuscript has been judged scientifically suitable for publication and will be formally accepted for publication once it meets all outstanding technical requirements.

Kind regards,

Leigh T. Stephenson

Academic Editor

PLOS ONE

Additional Editor Comments (optional):

Dear Dr Macauley,

Thanks for your patience w.r.t. the reviewing process. As you noted in your letter, the reviewers did take their time but I think you and they can be pleased with the result. Thank you for taking the time to respond to each of the reviewers' points in turn. The method employed by your group certainly has advantages and I hope that the technical solutions to the problems encountered by the initially ill-fitting puck have successfully remedied the observed shifting in the field desorption maps. I look forward to seeing this "in print".

Kind regards,

Leigh Stephenson (MPIE)
---

## [Editor Report · Acceptance letter]

8 Jan 2021

PONE-D-20-27616R1 

A versatile cryo-transfer system, connecting cryogenic focused ion beam sample preparation to atom probe microscopy 

Dear Dr. Macauley:

I'm pleased to inform you that your manuscript has been deemed suitable for publication in PLOS ONE. Congratulations! Your manuscript is now with our production department. 

Kind regards, 

on behalf of

Dr. Leigh T. Stephenson 

Academic Editor

PLOS ONE